# Trend of cancer mortality of the female reproductive system in China from 2005 to 2018 and prediction to 2035: A log-linear regression and Bayesian age-period- cohort analysis

Lili Xu[1‡], Ting Zhao[1], Xin-hua Wang[2], Guang-sheng Wu[3]*, Weixia Nong [4]*

**1** Department of General Medicine, The First Affiliated Hospital of Shihezi University, Shihezi, Xinjiang, China, **2** Department of Pathology, Shihezi People's Hospital, Shihezi, Xinjiang, China, **3** Department of Hematology, Friendship Hospital of Kazak Autonomous Prefecture, Yili, Xinjiang, China, **4** Department of Hematology, The First Affiliated Hospital of Shihezi University, Shihezi, Xinjiang, China

‡ The first author: Lili Xu.
* nwx_good@126.com (WN); hematology@126.com (GW)

## Abstract

### Objective

In recent years, the number of deaths from female reproductive system cancer in China has been continuously increasing, and there are relatively few studies on their mortality situation. This study aimed to analyze the mortality rate and its trend of the female reproductive system cancer in China from 2005 to 2018, to predict the situation until 2035, and to provide scientific basis for the prevention and control of the female reproductive system cancer in China.

### Methods

The mortality rate latest data of the female reproductive system cancer was obtained from the Chinese Cancer Registry Annual Report from 2008 to 2021. Using the Joinpoint regression model, the annual percent change (APC) and average annual percentage change (AAPC) were calculated to describe the time trend. Age-period-cohort models were constructed to analyze the effects of age, period, and cohort. The study predicted the situation up to 2035 using a Bayesian age-period-cohort model.

### Result

The age-standardized mortality rate (ASMR) of the female reproductive system cancer in China from 2005 to 2018 was 8.75/100,000, with rural areas being higher than urban areas. The mortality rate showed an upward trend from 2005 to 2018, with AAPC = 1.92% (95% *CI*: 1.68%, 2.16%). The growth trend in urban areas was more pronounced than that in rural areas, and the mortality risk was highest among the population aged 80–84 years. The period effect showed a trend of first increasing

**Data availability statement:** All relevant data are within the manuscript and its Supporting information files.

**Funding:** This research was supported by National Hematology Clinical Research Center Xinjiang production and construction Corps branch center (S2022EB517) and Correlation between CD5 expression and tumor-associated macrophage immunosuppression and disease prognosis in diffuse large B-cell lymphoma(2022ZD087). The funders had no role in study design, data collection and analysis, decision to publish, or preparation of the manuscript.

**Competing interests:** The authors declare that they have no known competing financial interests or personal relationships that could have appeared to influence the work reported in this paper.

and then decreasing from 2005 to 2018, with the highest risk in the population born between 1990 and 1994. The mortality rate of the female reproductive system cancer in China will increase from 9.96/100,000 (95% *CI*: 9.37/100,000, 10.54/100,000) in 2019 to 11.98/100,000 (95% *CI*: 3.64/100,000, 20.31/100,000) in 2035.

## Conclusion

The mortality rate of the female reproductive system cancer in China showed an upward trend from 2005 to 2018, and will continue to rise from 2019 to 2035. The female reproductive system cancer in China require continuous attention.

## Introudction

Cancer is a significant barrier to increasing global life expectancy, and have become the fourth leading cause of death among young people worldwide. It is also a major public health issue affecting the health and social development of residents in China [1].

Any pathological changes in the internal and external genitalia of the female reproductive system can lead to the development of cancers. According to the World Health Organization's Global Cancer Observatory database, in 2022, the highest mortality rate for female reproductive system cancer in China was cervical cancer, followed by ovarian cancer and endometrial cancer. The incidence age of cervical cancer is gradually becoming younger due to the widespread availability of screening technologies [2]. However, ovarian cancer, located deep within the pelvic cavity, has a concealed onset and nonspecific early clinical manifestations. The lack of effective early screening measures results in a higher mortality rate [3,4]. The prognosis for advanced endometrial cancer is poor, and improving the survival rate and prognosis for patients with mid-to-late-stage endometrial cancer remains one of the clinical challenges [5].

Improving prevention awareness and effectively controlling risk factors play a crucial role in reducing the burden of mortality and premature death from cancers, while early detection can reduce the likelihood of females dying from cancers such as cervical cancer. This study analyzed the trend in mortality rates from the female reproductive system cancer in China from 2005 to 2018, as well as the impact of age, period, and cohort, and made predictions up to 2035. The aim was to enhance the understanding and prevention of the female reproductive system cancer and promote measures to reduce the burden of these cancers.

## Materials and methods

### Data sources

The data on mortality rates of the female reproductive system cancer in China from 2005 to 2018 were derived from the Chinese Cancer Registry Annual Report from 2008 to 2021 [6–19]. To eliminate the effect of the age structure of the population on the level of death, it is necessary to calculate the age-standardized mortality

rate (ASMR), that is, the mortality rate calculated according to the age structure of a certain standard population. In this study, the standard population used was the population composition of the sixth national census in 2010 released by the National Bureau of Statistics of China [20].

Cases of the female reproductive system cancer are identified according to the International Classification of Diseases (ICD-10/C51-58) [21]. The urban-rural classification was based on the standards used in the Chinese Cancer Registry Annual Report, where urban areas were defined as prefecture-level regions and above, and rural areas as counties and county-level cities.

## Statistical analysis

**Joinpoint regression model.** Joinpoint regression primarily used linear and log-linear models to segmentally fit the time trend of mortality rate. In the analysis of population-based trend in cancer mortality, log-linear models are commonly selected [22]. This model utilizes a grid search method for fitting to establish all potential interval segment function joinpoints. Subsequently, a Monte Carlo permutation test was employed to determine the optimal number of joinpoints, leading to the construction of the best-fitting model.

The formula for the log-linear model is:

$$E[y/x] = e^{\beta_0 + \beta_1 x + \delta_1 (x - \tau_1)^+ + \cdots + \delta_k (x - \tau_k)}$$

Where, e denotes the base of the natural logarithm, k represents the number of joinpoints, $\tau_k$ is the unknown quantity of joinpoints. $\beta_0$ stands for the constant parameter, $\beta_1$ is the regression coefficient, $\delta_k$ indicates the slope of the k-th sub function. When $x - \tau_k > 0$, let $(x - \tau_1)^+ = x - \tau_k$; otherwise, $(x - \tau_1)^+ = 0$.

The period from 2005 to 2018 was divided into different sub-periods, with the annual percent change (APC) used to assess trend within each segment, and the average annual percent change (AAPC) used to evaluate the overall trend. A 95% confidence interval (CI) was employed to determine whether the trend was statistically significant. When the number of joinpoints was zero, APC = AAPC. When APC or AAPC>0, indicated an increasing trend within that time period, while a value less than zero indicated a decreasing trend.

The calculation formula for APC is:

$$APC = \left(e^{\beta_1} - 1\right) \times 100$$

The calculation formula for AAPC is:

$$AAPC = \left[\exp\left(\sum \omega_i \beta_i / \sum \omega_i\right) - 1\right] \times 100$$

Where, $\beta_1$ represents the regression coefficient, $\beta_i$ denotes the regression coefficient corresponding to each interval, and $\omega_i$ is the span of each interval (i.e., the number of years it encompasses).

**Age-period-cohort model.** The age-period-cohort model is based on a Poisson distribution and goodness-of-fit tests are used to identify the independent effects of age, period, and cohort factors. This approach allows for the estimation of coefficients for these three effects on the observed variable. To avoid potential unrecognizability issues due to the linear relationship among age, period, and cohort, the method of intrinsic estimators is employed to calculate the effect coefficients of age, period, and cohort [23]. Finally, by fitting a nested model of age, period, and cohort factors, criteria such as the Akaike Information Criterion and the Bayesian Information Criterion are used to determine whether these factors have a significant impact on the observed variable [24]. The formula for the model is as follows:

$$\log\left(\lambda_{apc}\right) = \alpha_a + \beta_p + \gamma_c + \varepsilon$$

Where, *α, β,* and *γ* represent the effects of age, period, and cohort, respectively, ε denotes the residuals. The age groups in this study were divided into one age group at intervals of five years as stated in the Chinese Cancer Registry Annual Report. The relative risk (RR) of mortality was used to describe trend changes, with RR < 1 indicating a decreased risk of death; RR > 1 indicates an increased risk of death.

**Bayesian age-period-cohort model.** The Bayesian age-period-cohort (BAPC) model utilizes three factors: age, period, and birth cohort, to collectively reflect numerous other influencing factors, thereby predicting mortality rates. The mortality rate for chronic diseases are relatively stable, especially for cancers, and this model is comparatively reliable for predicting such stable trend.

### Statistical analysis

Joinpoint regression analysis was conducted using the Joinpoint Regression Program version 5.0.2. The age-period-cohort model was constructed using an online web-based analysis tool [25]. Predictive analyses were performed with R software version 4.3.2. The level of significance was set at $P < 0.05$.

### Results

#### Mortality rate and its trend of the female reproductive system cancer in China from 2005 to 2018

From 2005 to 2018, the total number of deaths from the female reproductive system cancer in China was 166,480, with 91,158 cases (54.76%) in urban areas and 75,322 cases (45.24%) in rural areas. The overall crude mortality rate nation-wide was 11.36 per 100,000, with urban areas at 11.75 per 100,000 and rural areas at 10.93 per 100,000. The ASMR was 8.75 per 100,000, and the ASMR in urban areas was lower than that in rural areas ($\chi^2 = 6.19$, $P = 0.01$). Additionally, the urban-rural ratio of the crude mortality rate in 2005 was 1.16 (95% CI: 1.05, 1.27), which increased to 1.29 (95% CI: 1.22, 1.36) in 2010, and then decreased to 1.06 (95% CI: 1.04, 1.09) in 2018 (Table 1). It was further found that among the female reproductive system cancers, cervical uteri cancer had the highest ASMR (3.34 per 100,000) and placenta cancer had the lowest ASMR (0.01 per 100,000) (S1 File).

During 2005–2018, the ASMR for the female reproductive system cancer in China increased from 7.65 per 100,000 to 9.76 per 100,000, with an AAPC of 1.92% (95% CI: 1.68%, 2.16%). In urban areas, it increased from 7.57 per 100,000 to 9.83 per 100,000, with an AAPC of 2.19% (95% CI: 1.75%, 2.64%). In rural areas, it increased from 7.90 per 100,000 to 9.69 per 100,000, with an AAPC of 1.17% (95% CI: 0.38%, 1.96%). (Table 2, Fig 1)

#### Age-period-cohort model analysis

The impact of age on the mortality rate of the female reproductive system cancer from 2005 to 2018 was essentially consistent across the nation and in urban areas. In the 0–29 age group, the risk of developing the female reproductive system cancer increased slowly with age, and more rapidly in the 30–79 age group, peaking at 80–84 years. In rural areas, the risk of developing cancer of this system increased slowly with age in the 0–24 age group and more rapidly in the 25–84 age group.

In terms of period effects, the risk of death from the female reproductive system cancer at the national and urban levels showed an increasing trend from 2005 to 2010 and a decreasing trend from 2010 to 2015. In rural areas, the trend was the opposite, with the risk of death decreasing from 2005 to 2010 and increasing from 2010 to 2015.

Regarding cohort effects, the risk of death from the female reproductive system cancer was highest in the cohort born between 1990 and 1994, and then gradually declined. In urban areas, the risk of death showed an increasing trend, with a turning point in the cohort born between 1985 and 1989, and then gradually decreased. (Fig 2)

#### Prediction of mortality rate from the female reproductive system cancer in China from 2019 to 2035

From 2019 to 2035, the mortality rate of the female reproductive system cancer in China was predicted to show a gradual upward trend, increasing from 9.96 per 100,000 in 2019 (95% CI: 9.37/100,000, 10.54/100,000) to 11.98 per 100,000 by

**Table 1. Mortality rate of the female reproductive system cancer in China from 2005 to 2018 (1/100,000) and urban-rural ratio.**

| Year | Nationwide | | | Urban | | | Rural | | | Urban-rural ratio |
|------|--------|------|------|--------|------|------|--------|------|------|-------------------|
| | Number | Rate | ASMR | Number | Rate | ASMR | Number | Rate | ASMR | |
| 2005 | 2,353 | 8.68 | 7.65 | 1,801 | 9.01 | 7.57 | 552 | 7.77 | 7.90 | 1.16 (1.05, 1.27) |
| 2006 | 2,640 | 8.93 | 7.83 | 2,081 | 9.02 | 7.57 | 559 | 8.62 | 8.98 | 1.05 (0.95, 1.15) |
| 2007 | 2,768 | 9.36 | 8.08 | 2,090 | 9.48 | 7.79 | 677 | 9.00 | 9.15 | 1.05 (0.96, 1.15) |
| 2008 | 3,168 | 9.66 | 7.92 | 2,590 | 10.00 | 7.93 | 576 | 8.35 | 7.85 | 1.20 (1.09, 1.31) |
| 2009 | 4,191 | 9.92 | 8.41 | 2,989 | 10.49 | 8.48 | 1,200 | 8.74 | 8.17 | 1.20 (1.12, 1.28) |
| 2010 | 6,219 | 10.10 | 8.63 | 4,349 | 10.97 | 9.03 | 1,869 | 8.52 | 7.82 | 1.29 (1.22,1.36) |
| 2011 | 7,435 | 10.31 | 8.78 | 4,631 | 10.63 | 8.60 | 2,803 | 9.83 | 9.06 | 1.08 (1.03, 1.13) |
| 2012 | 10,217 | 10.46 | 8.82 | 5,530 | 11.08 | 8.85 | 4,686 | 9.80 | 8.78 | 1.13 (1.09, 1.18) |
| 2013 | 11,989 | 10.74 | 8.89 | 6,354 | 11.45 | 8.94 | 5,638 | 10.04 | 8.81 | 1.14 (1.10, 1.18) |
| 2014 | 15,810 | 11.13 | 9.14 | 8,432 | 11.77 | 9.23 | 7,382 | 10.49 | 9.04 | 1.12 (1.09, 1.16) |
| 2015 | 18,336 | 11.59 | 9.32 | 9,486 | 12.37 | 9.58 | 8,853 | 10.87 | 9.07 | 1.14 (1.11, 1.17) |
| 2016 | 23,112 | 12.30 | 9.78 | 12,442 | 13.00 | 10.07 | 10,668 | 11.57 | 9.45 | 1.12 (1.09, 1.15) |
| 2017 | 25,968 | 12.07 | 9.51 | 13,176 | 12.40 | 9.50 | 12,793 | 11.74 | 9.52 | 1.06 (1.03, 1.08) |
| 2018 | 32,275 | 12.53 | 9.76 | 15,206 | 12.94 | 9.83 | 17,061 | 12.18 | 9.69 | 1.06 (1.04, 1.09) |
| Total | 166,480 | 11.36 | 8.75 | 91,158 | 11.75 | 8.78 | 75,315 | 10.93 | 8.81 | 1.07 (1.07, 1.09) |

ASMR: age standardized mortality rate.

**Table 2. Trends in mortality rate of the female reproductive system cancer in China from 2005 to 2018 (%).**

| Index | Nationwide | Urban | Rural |
|-------|-----------|-------|-------|
| APC (95%CI) | 1.92 (1.68, 2.16) | 2.19 (1.75, 2.64) | 1.17 (0.38, 1.96) |
| t | 17.57 | 10.85 | 3.25 |
| P | <0.001 | <0.001 | 0.010 |
| AAPC (95%CI) | 1.92 (1.68, 2.16) | 2.19 (1.75, 2.64) | 1.17 (0.38, 1.96) |
| t | 17.57 | 10.85 | 3.25 |
| P | <0.001 | <0.001 | 0.010 |

AAPC: Annual average percentage change; APC: Annual percentage change; CI: Confidence interval.

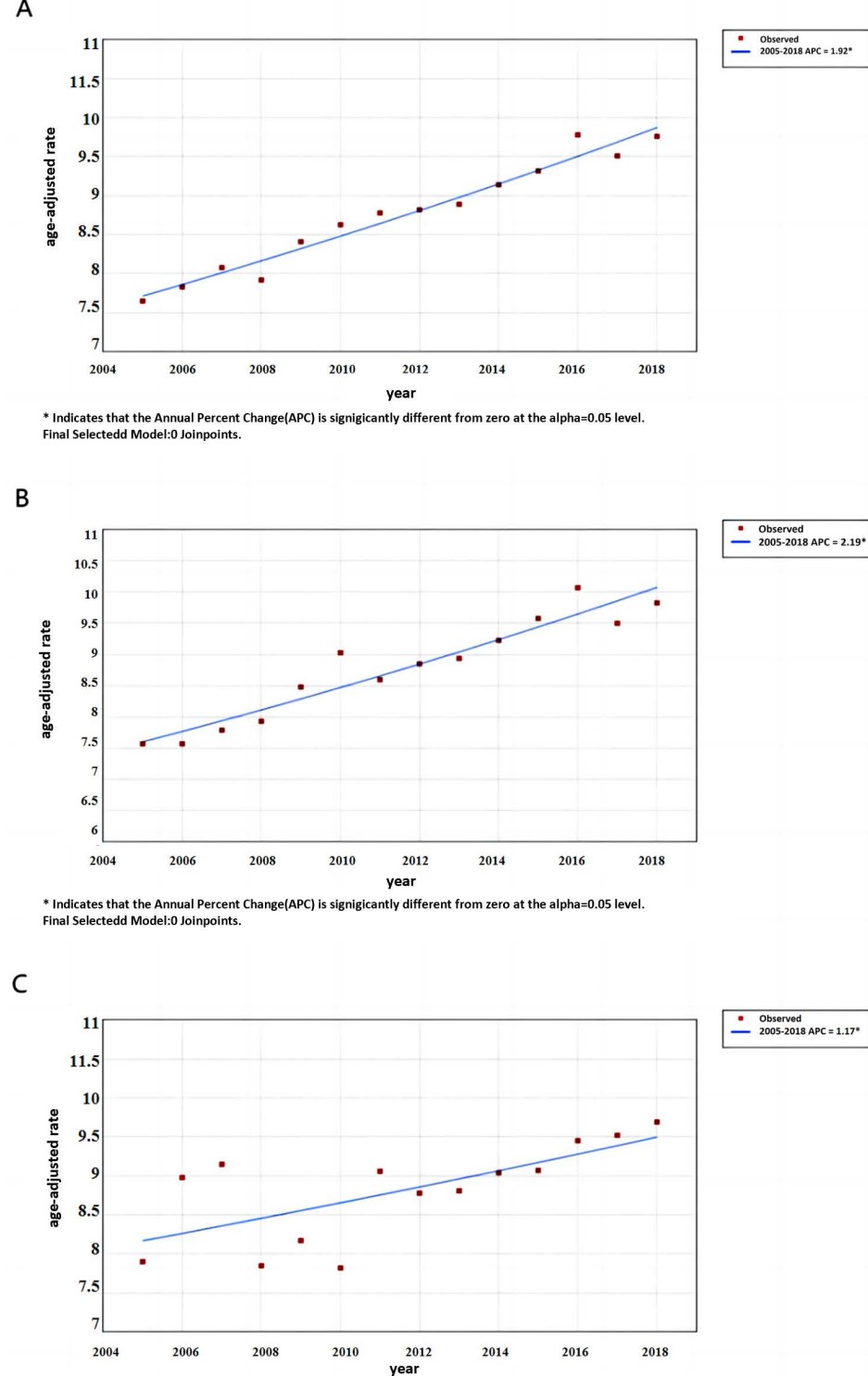

**A**

* Indicates that the Annual Percent Change(APC) is signigicantly different from zero at the alpha=0.05 level.
Final Selectedd Model:0 Joinpoints.

**B**

* Indicates that the Annual Percent Change(APC) is signigicantly different from zero at the alpha=0.05 level.
Final Selectedd Model:0 Joinpoints.

**C**

* Indicates that the Annual Percent Change(APC) is signigicantly different from zero at the alpha=0.05 level.
Final Selectedd Model:0 Joinpoints.

**Fig 1. Trends in the female reproductive system cancer mortality rate in China from 2005 to 2018 (A nationwide, B urban, C rural). APC: Annual percentage change.**

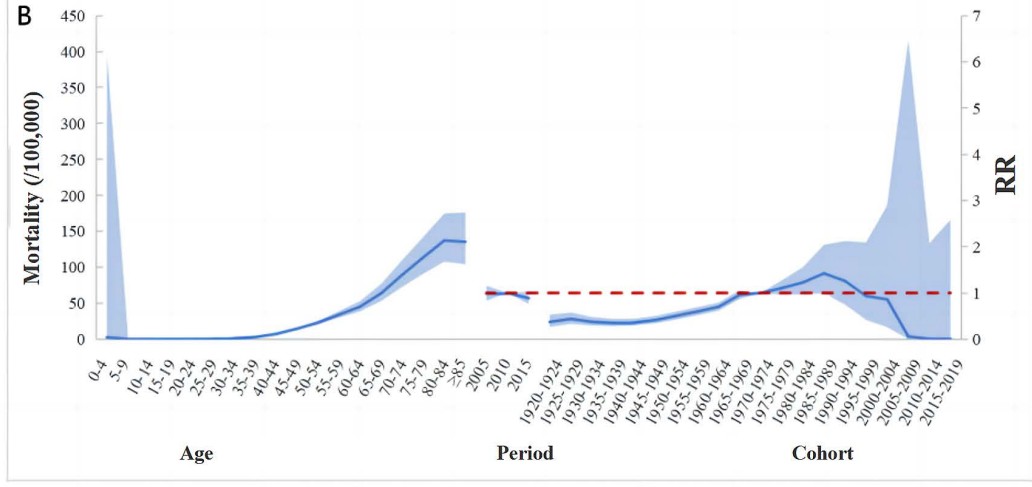

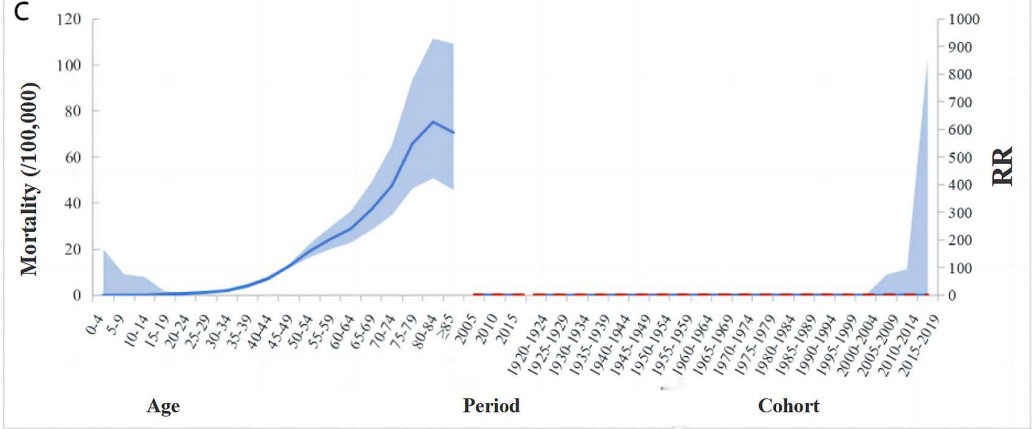

**Fig 2. Age-period-cohort effect s on the female reproductive system cancer mortality rate from 2005 to 2018 (A nationwide, B urban, C rural). RR: Risk ratio.**

2035 (95% CI: 3.64/100,000, 20.31/100,000). In urban areas, the rate was predicted to rise from 9.89 per 100,000 (95% CI: 8.96/100,000, 10.82/100,000) to 10.60 per 100,000 (95% CI: −2.25/100,000, 23.45/100,000). In rural areas, the rate was predicted to increase from 9.83 per 100,000 (95% CI: 9.08/100,000, 10.58/100,000) to 15.32 per 100,000 (95% CI: −5.15/100,000, 35.80/100,000). (Fig 3)

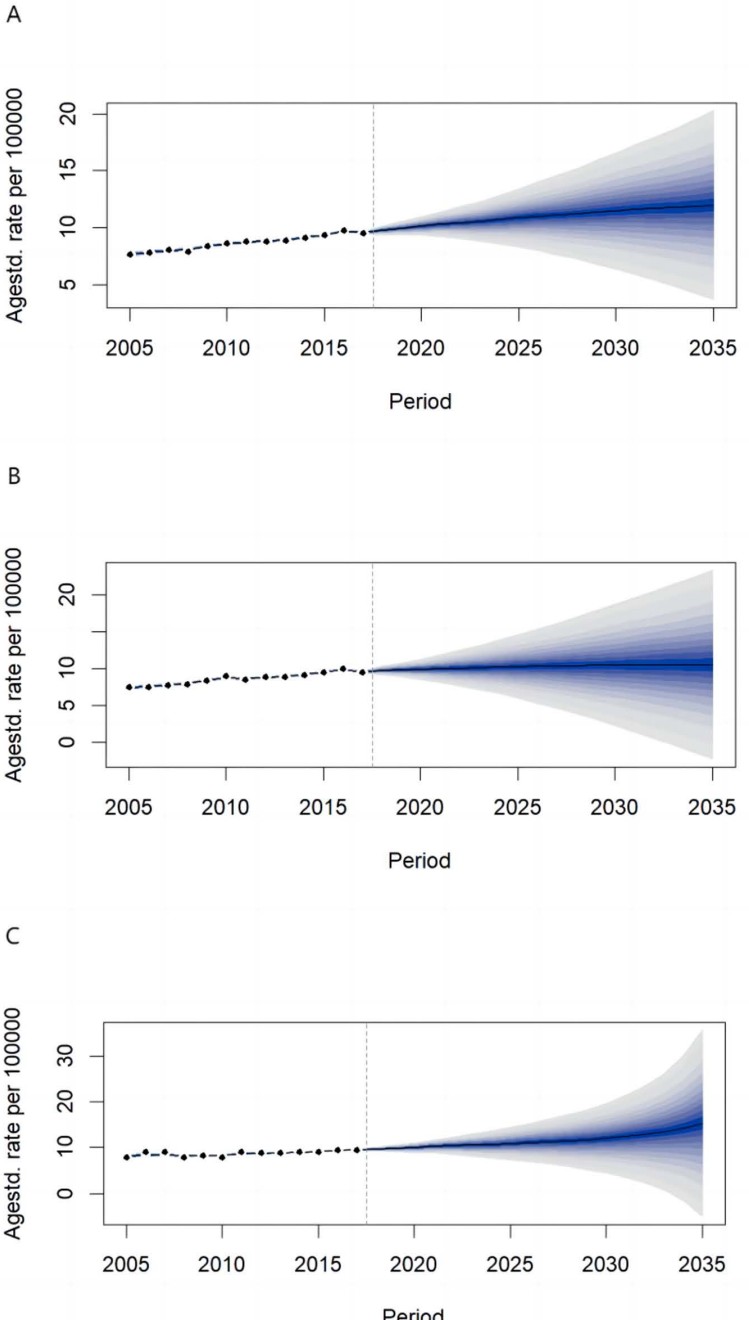

**Fig 3. Prediction of the female reproductive system cancer mortality rate in China from 2019 to 2035 (A nationwide, B urban, C rural).**

## Discussion

The mortality rate of the female reproductive system cancer was still at a high level in China. The reasons may include an aging population and unhealthy lifestyle habits. The aging population in China is one of the main drivers of the increasing cancer burden in recent years [26]. In addition, sedentary behavior can increase the risk of ovarian cancer, and prolonged sitting has become the norm in today's society [27]. At the same time, the rate of overweight and obesity among urban and rural residents in China continues to rise, with more than half of the adult residents being overweight or obese [28]. Studies have shown that obesity has a negative impact on the prognosis and treatment outcomes of the female reproductive system cancer [29]. Obesity and a higher percentage of visceral fat also reduce the survival rate of endometrial cancer [30]. Additionally, exposure to PM2.5 is associated with a shorter survival period in patients with cervical cancer [31]. From 2000 to 2020, the concentration of PM2.5 in China showed a trend of first increasing and then decreasing, peaking in 2013 (34.81 micrograms per cubic meter), which was more than six times the guideline level of the World Health Organization in 2021 [32]. In summary, long-term exposure to these risk factors may contribute to the continuous increase in the mortality rate of the female reproductive system cancer.

This study also predicted that the ASMR of the female reproductive system cancer in China will continue to rise until 2035. The 2024 edition of the Global Obesity Map released by the World Obesity Federation indicated that the rate of overweight and obesity among Chinese urban and rural residents will trend upwards in the future [33]. While PM2.5 levels were expected to decline in the coming years, the health benefits of improved air quality may be offset by changes in the age structure of the Chinese population.

The ASMR for the female reproductive system cancer was higher in rural areas. The "2020 National Aging Development Report" indicated that the level of aging in China's rural areas was significantly higher than that in urban areas, which may lead to a higher mortality rate of female reproductive system cancer in rural areas compared to urban areas [34]. The Chinese Center for Disease Control and Prevention reported that the rate of increase in overweight and obesity was faster in rural areas of our country. Furthermore, compared to urban patients, rural patients with cancer face inequities in payment burdens and health insurance benefits, resulting in lower utilization rates of healthcare services, including screening and treatment for cancers [35]. The combined effect of these factors may contribute to the higher mortality rate of the female reproductive system cancer in rural areas compared to urban areas.

The cohort effect results indicated that individuals born between 1990 and 1994 had the highest risk of death from the female reproductive system cancer, with a subsequent downward trend. These changes may be associated with the aging of the Chinese population and the implementation of preventive measures such as the "two cancers screening" initiative and environmental protection during this period. China places high importance on public health, having implemented the most stringent air pollution control action plan in history in 2013. In 2023, the Chinese government issued a notice on the "Action Plan for Continuous Improvement of Air Quality," setting a target to reduce PM2.5 concentrations in cities at the prefectural level and above by 10% compared to 2020 by 2025, and to control the proportion of days with severe pollution to within 1% [36]. The "Healthy China 2030" Planning Outline, launched in 2016, provides a historic opportunity to develop a national strategy to address the issue of obesity. The "Action Plan for Accelerating the Elimination of Cervical Cancer (2023-2030)" issued by the National Health Commission has promoted the "two cancers" screening services for females, early diagnosis and treatment of cervical cancer, and the continuous standardization of diagnostic and treatment technologies.

There are some limitations in this study. The data in this study were collected from Chinese Cancer Registry Annual Report, and the sample representativeness may be insufficient due to the coverage. Also, the fact that these data were obtained from the Chinese population limits the generalizability of direct rate comparisons with other countries or globally. Additionally, this study did not analyze influencing factors of the female reproductive system cancer, so no causal conclusions can be drawn.

## Conclusion

In summary, the ASMR for the female reproductive system cancer in China showed an overall upward trend from 2005 to 2018 and was predicted to continue rising from 2019 to 2035. The gap in mortality trend between urban and rural areas was continuously narrowing. At the same time, the risk of death from the female reproductive system cancer increased with age. We should continue to focus on the mortality trend and influencing factors of the female reproductive system cancer, maintain the emphasis on the prevention and control of these cancers, and improve the overall health and lifestyle of the population.

## Supporting information

**S1 File. The mortality rates of eight types of the female reproductive system cancer in China from 2005 to 2018.**
(DOC)

**S2 File. The original data on female reproductive system cancer mortality sourced from Chinese Cancer Registry Annual Report.**
(XLSX)

## Acknowledgments

We thank the Chinese Cancer Centers for their time and effort in preparing these publicly available data.

## Author contributions

**Conceptualization:** Lili Xu, Weixia Nong.

**Formal analysis:** Lili Xu, Ting Zhao, Xin-hua Wang, Guang-sheng Wu, Weixia Nong.

**Funding acquisition:** Weixia Nong.

**Methodology:** Lili Xu, Ting Zhao, Xin-hua Wang, Guang-sheng Wu, Weixia Nong.

**Project administration:** Weixia Nong.

**Software:** Lili Xu, Ting Zhao, Xin-hua Wang, Guang-sheng Wu, Weixia Nong.

**Supervision:** Weixia Nong.

**Writing – original draft:** Lili Xu, Weixia Nong.

**Writing – review & editing:** Lili Xu, Weixia Nong.

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
