## [Decision Letter · Decision Letter 0]

Dear Dr.  Nong,

Thank you for submitting your manuscript to PLOS ONE. After careful consideration, we feel that it has merit but does not fully meet PLOS ONE’s publication criteria as it currently stands. Therefore, we invite you to submit a revised version of the manuscript that addresses the points raised during the review process.

We look forward to receiving your revised manuscript.

Kind regards,

Giuseppe Di Martino

Academic Editor

PLOS ONE

Journal Requirements:

Reviewers' comments:

Reviewer's Responses to Questions

**Comments to the Author**

1. Is the manuscript technically sound, and do the data support the conclusions?

Reviewer #1: Yes

Reviewer #2: Partly

2. Has the statistical analysis been performed appropriately and rigorously?

Reviewer #1: Yes

Reviewer #2: Yes

3. Have the authors made all data underlying the findings in their manuscript fully available?

Reviewer #1: Yes

Reviewer #2: Yes

4. Is the manuscript presented in an intelligible fashion and written in standard English?

Reviewer #1: Yes

Reviewer #2: Yes

Reviewer #1: Thank you for asking me to review the manuscript. For ease and compliance for editing, I had to convert and download as MSM instead of PDF.

The manuscript is well written and the information provided is useful to the scientific world. However I wish to observe and suggest as follows:

1. Title: The title may have to be edited as shown in the annotations in the attached suggestion for clarity.

2. Main text: Other errors and suggestions as attached in the annotations in the attachment.

3. Discussion: The first paragraph here is essentially a repetition of the results, so should either be expunged or rephrased.

Reviewer #2: 1.Review the manuscript for some typos.

2. Considering C51-58 as a whole - Since these involve topographies with different risk factors, incidence, and mortality, I suggest that the authors at least present a table with showing rates of each.

3. For the first mortality rates presented in results, Are these crude or age-standardized rates?

4. How was the ASMR calculated and which was the reference population? Describe in methods.

5. Describe in Methods how was the age groups achieved.

6. The authors perform a discussion of risk of individual female reproductive malignancies, but their research did not approach individual topographies. I suggest, some data of individual topographies be given.

7. In limitations, it is said that this is a cross-sectional study. So, I suggest revising.

**Do you want your identity to be public for this peer review?** For information about this choice, including consent withdrawal, please see our Privacy Policy

Reviewer #1: **Yes: ** Matthias Gabriel Abah

Reviewer #2: No

---

## [Author Response · Author response to Decision Letter 1]

5 May 2025

PLOS ONE

April 2025

PONE-D-25-02080

Trend of cancer mortality of the female reproductive system in China from 2005 to 2018 and prediction to 2035: a Log-linear regression and Bayesian age-period- cohort analysis

Dear editor and reviewers,

We appreciate editor and reviewers very much for their positive and constructive comments and suggestions on our manuscript entitled “Trend of cancer mortality of the female reproductive system in China from 2005 to 2018 and prediction to 2035: a Log-linear regression and Bayesian age-period-cohort analysis” (Manuscript ID: PONE-D-25-02080). Those comments are all valuable and very helpful for revising and improving our paper, as well as the important guiding significance to our researches. Based on these comments and suggestions, we have made careful modification on the original manuscript.

On the separate pages, we provided our response to the comments and suggestions, point by point, and highlighted the changes in the marked copy of the revision. We hope that our revision will be approved by the experts and reviewed favorably.

Sincerely,

Weixia Nong, MD

Reviewer #1: Thank you for asking me to review the manuscript. For ease and compliance for editing, I had to convert and download as MSM instead of PDF.

The manuscript is well written and the information provided is useful to the scientific world. However I wish to observe and suggest as follows:

1. Title: The title may have to be edited as shown in the annotations in the attached suggestion for clarity.

Response Thank you for your reasonable suggestions. But we didn't find the attachment. Could you please send it again.

2. Main text: Other errors and suggestions as attached in the annotations in the attachment.

Response Thank you for your careful review. But we didn't find the attachment. Could you please send it again.

3. Discussion: The first paragraph here is essentially a repetition of the results, so should either be expunged or rephrased.

Response Thank you for your reasonable suggestions. We have deleted.

Reviewer #2:

1.Review the manuscript for some typos.

Response Thank you for your careful review. We have checked and modified.

2. Considering C51-58 as a whole - Since these involve topographies with different risk factors, incidence, and mortality, I suggest that the authors at least present a table with showing rates of each.

Response Thank you for your reasonable advice. We have added the mortality rates of eight female reproductive system cancers in the Supplementary Material (Table S1-S8), and added description in the results: “It was further found that among the female reproductive system cancers, cervical uteri cancer had the highest ASMR (3.34 per 100,000) and placenta cancer had the lowest ASMR (0.01per 100,000) (Table S1-S8).”

3. For the first mortality rates presented in results, Are these crude or age-standardized rates?

Response Thank you for your careful review. The first mortality rate that emerged in the results was the crude mortality rate, and the age standardized mortality rate was expressed by ASMR.

4. How was the ASMR calculated and which was the reference population? Describe in methods.

Response Thank you for your reasonable suggestions. We have added in the methods section: “To eliminate the effect of the age structure of the population on the level of death, it is necessary to calculate the age-standardized mortality rate (ASMR), that is, the mortality rate calculated according to the age structure of a certain standard population. In this study, the standard population used was the population composition of the sixth national census in 2010 released by the National Bureau of Statistics of China(20).”

5. Describe in Methods how was the age groups achieved.

Response Thank you for your reasonable suggestions. We have supplemented in the methods section: “The age groups in this study were divided into one age group at intervals of five years as stated in the Chinese Cancer Registry Annual Report.”

6. The authors perform a discussion of risk of individual female reproductive malignancies, but their research did not approach individual topographies. I suggest, some data of individual topographies be given.

Response Thank you for your reasonable request. We have added the mortality rates of eight female reproductive system cancers in the Supplementary Material (Table S1-S8).

7. In limitations, it is said that this is a cross-sectional study. So, I suggest revising.

Response Thank you for your reasonable suggestions. We have deleted.

---

## [Decision Letter · Decision Letter 1]

Dear Dr. Nong,

Thank you for submitting your manuscript to PLOS ONE. After careful consideration, we feel that it has merit but does not fully meet PLOS ONE’s publication criteria as it currently stands. Therefore, we invite you to submit a revised version of the manuscript that addresses the points raised during the review process.

Your study provides valuable insights into the mortality trends of female reproductive system cancers in China and employs appropriate statistical methodologies for trend analysis and prediction. The work addresses an important public health issue and will contribute meaningfully to the epidemiological literature. However, minor revisions are required before final acceptance

Required revisions:

The manuscript requires a complete revision of academic writing to enhance clarity, coherence, and overall readability. Please ensure the writing style aligns with scholarly standards.

**Discussion - Reference Citation:** In the second paragraph of the discussion, you mention "the 2024 edition of the Global Obesity Map released by the World Obesity Federation." Please provide a complete and proper reference citation for this source in your reference list.**Discussion - Third Paragraph:** The sentence "These reasons may be due to the increasing trend in the mortality rate of the female reproductive system cancer in urban areas was more pronounced than in rural areas, leading to a narrowing of the mortality gap between urban and rural areas" requires revision for clarity and grammatical correctness. Please, also be mindful of potential ecological fallacy when making causal inferences from population-level data.**Methodological Note:** Please acknowledge in your limitations section that the reference population used was China's population, which limits the generalizability of direct rate comparisons to other countries or global contexts.

This decision is based on one reviewer's assessment due to the unavailability of the second reviewer's opinion. The revisions requested are minor and primarily focus on language clarity, proper citation, and methodological transparency.

We look forward to receiving your revised manuscript.

Kind regards,

Yordanis Enríquez Canto, Ph.D.

Academic Editor

PLOS ONE

Journal Requirements:

Reviewers' comments:

Reviewer's Responses to Questions

**Comments to the Author**

Reviewer #2: All comments have been addressed

2. Is the manuscript technically sound, and do the data support the conclusions?

Reviewer #2: Yes

3. Has the statistical analysis been performed appropriately and rigorously?

Reviewer #2: Yes

4. Have the authors made all data underlying the findings in their manuscript fully available?

Reviewer #2: Yes

5. Is the manuscript presented in an intelligible fashion and written in standard English?

Reviewer #2: Yes

Reviewer #2: -In the first paragragh of introduction review the verbal tense following "Cancer... has.

-Only a comment: the reference population used was China's, so worldwide comparisons of rates is not possible.

-2nd. paragraph of discussion: provide a reference for the 2024 edition of the Global Obesity Map.

-The last phrase of 3rd. paragraph in Discussion: "These reasons may be due to the increasing trend..." needs to be rewritten, probably the aforementioned factors may lead to the increasing mortality. However, beware of the so called "ecological fallacy".

**Do you want your identity to be public for this peer review?** For information about this choice, including consent withdrawal, please see our Privacy Policy

Reviewer #2: No

---

## [Author Response · Author response to Decision Letter 2]

1 Jul 2025

PLOS ONE

June 2025

PONE-D-25-02080

Trend of cancer mortality of the female reproductive system in China from 2005 to 2018 and prediction to 2035: a Log-linear regression and Bayesian age-period- cohort analysis

Dear editor and reviewers,

We appreciate editor and reviewers very much for their positive and constructive comments and suggestions on our manuscript entitled “Trend of cancer mortality of the female reproductive system in China from 2005 to 2018 and prediction to 2035: a Log-linear regression and Bayesian age-period-cohort analysis” (Manuscript ID: PONE-D-25-02080). Those comments are all valuable and very helpful for revising and improving our paper, as well as the important guiding significance to our researches. Based on these comments and suggestions, we have made careful modification on the original manuscript.

On the separate pages, we provided our response to the comments and suggestions, point by point, and highlighted the changes in the marked copy of the revision. We hope that our revision will be approved by the experts and reviewed favorably.

Sincerely,

Weixia Nong, MD

Required revisions:

The manuscript requires a complete revision of academic writing to enhance clarity, coherence, and overall readability. Please ensure the writing style aligns with scholarly standards.

Response We reviewed and improved the full text again to make the expression of this article clearer and more in line with academic standards. Thank you very much to the editors and experts for their careful review and valuable comments.

1.Discussion - Reference Citation: In the second paragraph of the discussion, you mention "the 2024 edition of the Global Obesity Map released by the World Obesity Federation." Please provide a complete and proper reference citation for this source in your reference list.

Response Thank you for your reasonable suggestions. We have supplemented the references.

2.Discussion - Third Paragraph: The sentence "These reasons may be due to the increasing trend in the mortality rate of the female reproductive system cancer in urban areas was more pronounced than in rural areas, leading to a narrowing of the mortality gap between urban and rural areas" requires revision for clarity and grammatical correctness. Please, also be mindful of potential ecological fallacy when making causal inferences from population-level data.

Response Thank you for your reasonable suggestions. We strongly agree with this statement. We also recognize that such an explanation is inappropriate, so we have deleted it.

3.Methodological Note: Please acknowledge in your limitations section that the reference population used was China's population, which limits the generalizability of direct rate comparisons to other countries or global contexts.

Response Thank you for your reasonable suggestions. We have added: “ Also, the fact that these data were obtained from the Chinese population limits the generalizability of direct rate comparisons with other countries or globally.”

---

## [Decision Letter · Decision Letter 2]

Trend of cancer mortality of the female reproductive system in China from 2005 to 2018 and prediction to 2035: a Log-linear regression and Bayesian age-period- cohort analysis

PONE-D-25-02080R2

Dear Dr. Nong,

We’re pleased to inform you that your manuscript has been judged scientifically suitable for publication and will be formally accepted for publication once it meets all outstanding technical requirements.

Kind regards,

Yordanis Enríquez Canto, Ph.D.

Academic Editor

PLOS ONE

Additional Editor Comments (optional):

Reviewers' comments:

Reviewer's Responses to Questions

**Comments to the Author**

Reviewer #2: All comments have been addressed

2. Is the manuscript technically sound, and do the data support the conclusions?

Reviewer #2: Yes

3. Has the statistical analysis been performed appropriately and rigorously?

Reviewer #2: Yes

4. Have the authors made all data underlying the findings in their manuscript fully available?

Reviewer #2: Yes

5. Is the manuscript presented in an intelligible fashion and written in standard English?

Reviewer #2: Yes

Reviewer #2: I have no additional points since the authors have addressed them all. I understand the manuscript is now ready to follow the editors commands.

**Do you want your identity to be public for this peer review?** For information about this choice, including consent withdrawal, please see our Privacy Policy

Reviewer #2: **Yes: ** Carlos Anselmo Lima

---

## [Editor Report · Acceptance letter]

PONE-D-25-02080R2

PLOS ONE

Dear Dr. Nong,

I'm pleased to inform you that your manuscript has been deemed suitable for publication in PLOS ONE. Congratulations! Your manuscript is now being handed over to our production team.

Kind regards,

on behalf of

Prof. Yordanis Enríquez Canto

Academic Editor

PLOS ONE